evolution, biochemistry, microbiology

autocatalytic networks, origin of metabolism, biochemical evolution, origin of life, methanogens, acetogens

**Author for correspondence:**
Joana C. Xavier
e-mail: xavier@hhu.de

# Autocatalytic chemical networks at the origin of metabolism

Joana C. Xavier[1], Wim Hordijk[2], Stuart Kauffman[3], Mike Steel[4] and William F. Martin[1,5]

[1]Institut für Molekulare Evolution, Heinrich Heine Universität, 40225 Düsseldorf, Germany
[2]Konrad Lorenz Institute for Evolution and Cognition Research, 3400 Klosterneuburg, Austria
[3]Institute for Systems Biology, Seattle, WA 98109-5263, USA
[4]Biomathematics Research Centre, University of Canterbury, Christchurch 8041, New Zealand
[5]Instituto de Tecnologia Química e Biológica, Universidade Nova de Lisboa, 2780-157 Oeiras, Portugal

JCX, 0000-0001-9242-8968

Modern cells embody metabolic networks containing thousands of elements and form autocatalytic sets of molecules that produce copies of themselves. How the first self-sustaining metabolic networks arose at life's origin is a major open question. Autocatalytic sets smaller than metabolic networks were proposed as transitory intermediates at the origin of life, but evidence for their role in prebiotic evolution is lacking. Here, we identify reflexively autocatalytic food-generated networks (RAFs)—self-sustaining networks that collectively catalyse all their reactions—embedded within microbial metabolism. RAFs in the metabolism of ancient anaerobic autotrophs that live from $H_2$ and $CO_2$ provided with small-molecule catalysts generate acetyl-CoA as well as amino acids and bases, the monomeric components of protein and RNA, but amino acids and bases without organic catalysts do not generate metabolic RAFs. This suggests that RAFs identify attributes of biochemical origins conserved in metabolic networks. RAFs are consistent with an autotrophic origin of metabolism and furthermore indicate that autocatalytic chemical networks preceded proteins and RNA in evolution. RAFs uncover intermediate stages in the emergence of metabolic networks, narrowing the gaps between early Earth chemistry and life.

## 1. Introduction

Cells are autocatalytic in that they require themselves for reproduction. The origin of the first cells from the elements on the early Earth roughly 4 billion years ago [1–4] must have been stepwise. The nature of autocatalytic systems as intermediate states in that process is of interest. Autocatalytic sets are simpler than cellular metabolism and produce copies of themselves if growth substrates for food and a source of chemical energy for thermodynamic thrust are provided [5–7]. In theory, sets of organic molecules should be able to form autocatalytic systems [8–12], which, if provided with a supply of starting 'food' molecules, can emerge spontaneously and proliferate via constraints imposed by substrates, catalysts, or thermodynamics [13]. Autocatalytic sets have attracted considerable interest as transitory intermediates between chemical systems and genetically encoded proteins at the origin of life [13–17]. Preliminary studies have shown that coenzymes are often required for their own synthesis and are therefore replicators with autocatalytic properties [18]. However, autocatalytic networks have not been identified in non-enzymatic metabolic networks so far, and evidence for their existence during prebiotic evolution is lacking.

Of special interest for metabolic evolution are a class of mathematical objects called reflexively autocatalytic food-generated networks—RAFs—in which each reaction is catalysed by a molecule from within the network and all molecules can be produced from a set of food molecules by the network itself [19]. RAFs

are a precisely defined type of autocatalytic set, for their emergence from this ambient food set. Other autocatalytic networks not food-generated (sometimes called pseudo-RAFs) have the property of persistence but not of emergence from a food supply. In a related class of objects named constructively autocatalytic food-generated networks—CAFs [20], catalysts must either be present in the food set or produced before their first requirement. By contrast, RAFs impose that all necessary catalysts need to be produced by the network at some point, but not necessarily at the first time they are required. This feature models the emergence of specificity, speed, and efficiency in autocatalysis. Depending on the order of non-catalysed reactions, different routes can exist for the formation of an RAF. By depending on food, and representing the emergence of catalytic specificity, RAFs are an appropriate model for the origin of metabolism. Small chemical systems resembling RAFs have been constructed in the laboratory [7,21–23], although still far from the scale of cellular metabolism, which is composed of thousands of reactions. Modern cellular metabolism is enzyme–based, but greater than 60% of enzyme mechanisms described to date involve one or more cofactors [24] and 40% of all proteins crystallised have a bound metal relevant to their function [25]. RAFs can thus be identified in modern metabolism [17] by attributing the catalysis of enzymes to their metals and cofactors in prebiotic evolution [26–31], generalizing the well-known observation that native metals [3,32–36], flavins [37], pyridoxal 5′-phosphate [38,39], S-adenosyl methionine (SAM) [40], NAD [41], CoA [42], thiamine diphosphate [43], folates [44], and other cofactors [26,30,45] can themselves perform catalysis in the absence of enzymes. Also relevant to the inference of RAFs within metabolic networks are the numerous non-enzymatic, spontaneous reactions that are known to occur in metabolism [31,46]. If autocatalytic chemical networks antedate genetically encoded proteins, cofactor-dependent RAFs might have been involved and, if so, should have left evidence for their existence in modern metabolic networks.

In search of RAFs, we investigated different levels of ancient metabolism preserved in modern cells. Starting with the biosphere level of the KEGG database, we first removed all eukaryote-specific reactions, and then peeled back one more layer of time by examining anaerobic metabolism. The detection of a large RAF in anaerobic prokaryotic metabolism prompted us to ask whether RAFs are also preserved in the metabolism of ancient anaerobic autotrophs that trace to the last universal common ancestor, LUCA [47]. As far back as we could look in metabolic evolution, RAFs were found. They were found in the metabolism of the acetogenic bacterium *Moorella thermoacetica* and the methanogenic archaeon *Methanococcus maripaludis*, lineages thought to be primitive as they live on the simplest source of carbon and energy known, the $H_2$-$CO_2$ redox couple [1,5,29,47–49], they assimilate geochemically generated carbon species [50,51], they generate ATP from $CO_2$ fixation [5], their core bioenergetic reactions occur abiotically in hydrothermal vents [48,52], and under laboratory conditions [3], their ecology and gene trees link them to LUCA [53], and they still inhabit primordial habitats within the crust today [54]. The RAFs of the methanogen and the acetogen furthermore intersect in a primordial network that generates amino acids, nucleosides, and acetyl-CoA from a starting set of simple food molecules, shedding light on the nature of autocatalytic networks that existed before the first cells arose from the elements on the early Earth.

# 2. Results
## (a) Two-thirds of global prokaryotic metabolism can be annotated with small-molecule catalysis

In search of RAFs in 4-billion-year-old metabolism, we started from all 10 828 KEGG reactions and purged the set of non-primordial reactions in two pruning steps. First, looking at the 8352 reactions assigned to enzymes, we removed reactions that occur only in eukaryotes. Such reactions are unlikely to be primordial, because eukaryotes arose less than 2 billion years ago [2]. Second, from the resulting set we excluded $O_2$-dependent reactions, because $O_2$ is a product of cyanobacterial photosynthesis, which arose about 2.4 billion years ago [55]. These pruning steps left 5847 enzyme-associated reactions, 66% of which involve at least one cofactor, and the remaining 33% were assigned an operational catalyst named 'peptide' which is attributed to reactions catalysed by enzymes to which no cofactor or metal is currently associated in the databases. From the initial set of 10 828 reactions, we identified, retrieved, and added to the previous set 147 spontaneous reactions, generating a global network comprising 5994 reactions and 5723 metabolites (electronic supplementary material, figure S1 and dataset S1A). The cofactors involved in this ancient anaerobic network are distributed among the six different enzyme commission (EC) classes as shown in figure 1. Cofactors can be both catalysts and substrates or products in reactions (for example, NAD) or just catalysts (metals). Metal catalysis is widespread across all classes of metabolism, and NADs dominate the oxido-reductase reactions. The resulting prokaryotic $O_2$-independent network contains 70% of all 8352 reactions associated with enzymes before removal of $O_2$-dependent and eukaryote-specific reactions, indicating that a great part of metabolism was invented in the anaerobic world [56].

## (b) Autocatalysis in global metabolism expands with a small set of cofactors

The largest possible RAFs (maxRAFs) in a network are of interest because they represent its largest component of autocatalytic complexity. Figure 2a shows a schematic of an RAF within a metabolic network. The RAF algorithm, first introduced in [19] and refined in [57] and [58], starts with the full reaction network, and iteratively removes reactions that fail to have all their reactants and at least one catalyst able to be produced from the food set via the current reaction set. For this, the 'closure of the food set' needs to be computed at each iteration (electronic supplementary material, Methods). The algorithm ends when no more reactions can be removed. If the remaining set of reactions is non-empty, it comprises the maxRAF, i.e. the union of all possible RAFs in the original reaction network. If the remaining set is empty, there was no RAF in the original network.

The maxRAFs in the global prokaryotic $O_2$-independent network were identified for different food sets, that is, molecules provided by the environment (electronic supplementary material, table S1). An inorganic food set containing $H_2O$, $H_2$, $H^+$, $CO_2$, $CO$, $PO_4^{3-}$, $SO_4^{2-}$, $HCO_3^-$, $P_2O_7^{4-}$, S, $H_2S$, $NH_3$, $N_2$, all metals, FeS clusters and other metal clusters, a generalist acceptor, donor, and metal produced a minute maxRAF with eight reactions linking ammonia, carbon, and sulfide transformations. The addition

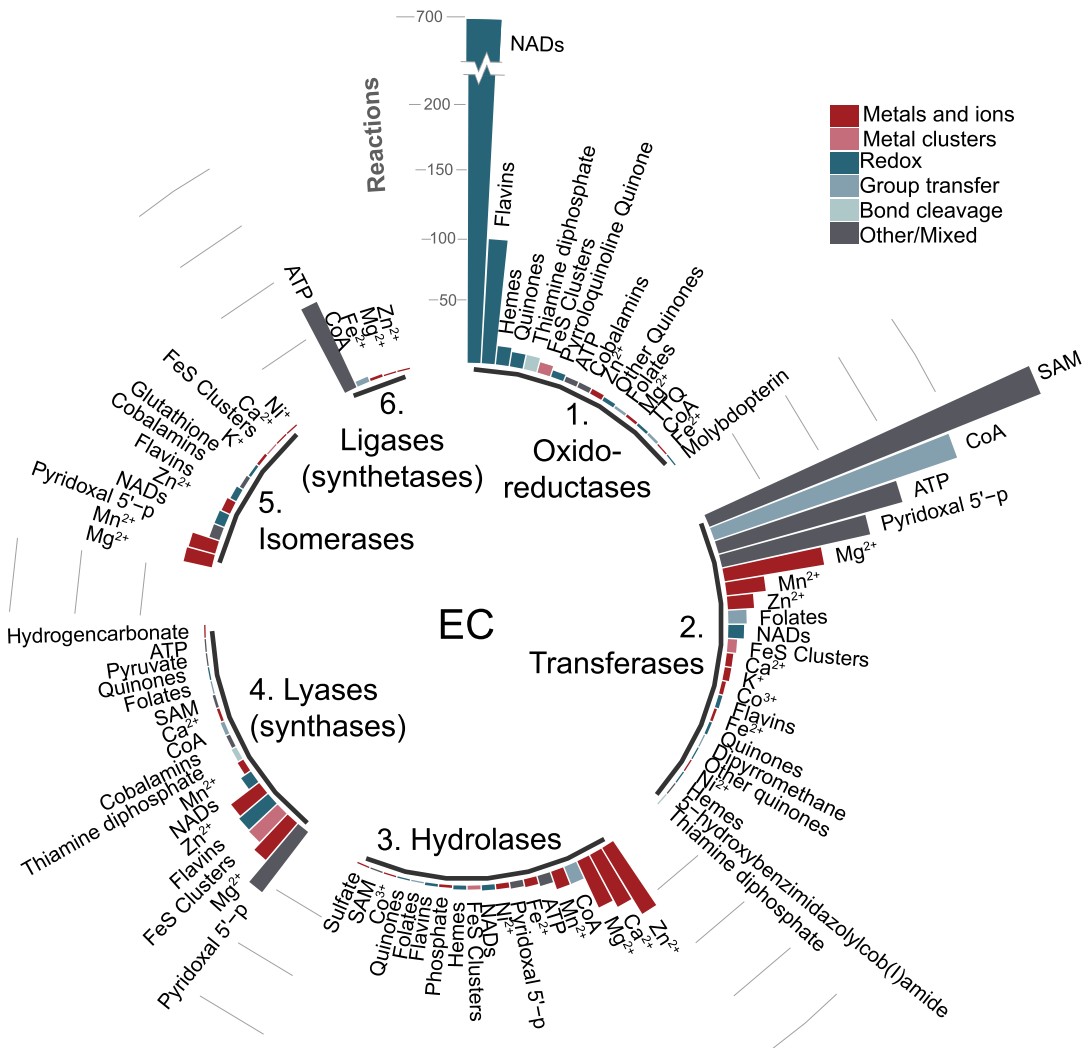

**Figure 1.** Catalysts in global oxygen-independent prokaryotic metabolism. The catalysis-annotated network separated by enzyme commission (EC) classes with the corresponding cofactors for each. Cofactors are grouped (legend, top right) according to their function in catalysis. NADs, Cobalamins, Folates, Flavins, and Quinones are each a group of equivalent catalysts with common properties and common biosynthetic pathways (for example, NADs stands for NAD(P)(H); all pooling reactions are detailed in electronic supplementary material, dataset S1A).

of formate, methanol, acetate, and pyruvate, which are central metabolites with experimental evidence for synthesis from $CO_2$ and metals [3], doubles the maxRAF size to 16 reactions. In principle, the addition of organic cofactors (electronic supplementary material, table S1) to the food set should generate larger maxRAFs. Sequential addition of the eight most frequent cofactors identified in the LUCA's proteins [47] to the metal–$CO_2$ food set expanded the maxRAF from 16 to 914 reactions (figure 2b). The addition of all cofactors germane to the anaerobic network generates a maxRAF with 1335 reactions spanning 25% of the starting anaerobic network. Sequential addition of the eight compounds that were most frequent in that maxRAF, to the metal–$CO_2$ food set, expands the maxRAF from 16 to 1066 reactions, whereas sequential addition of the five compounds with the greatest impact (upon removal from the food set) on anaerobic maxRAF size followed by the three most frequent in the largest maxRAF yields a final maxRAF of 1248 reactions (figure 2b). These results indicate that RAFs can grow in size through sequential incorporation of organic cofactors (figure 2b). RAFs can thus provide structure, contingency, increasing complexity, and direction to interactions among molecule food sets, given a sustained geochemical source of carbon, energy, and electrons.

## (c) Autocatalytic networks point to an early autotrophic metabolism

If autocatalytic sets were instrumental at the origin of metabolism [13], lineages with a physiology very similar to that of the first cells should harbour the most ancient RAFs.

Several lines of evidence indicate that methanogens and acetogens reflect the ancestral state of microbial physiology in the bacteria and archaea, respectively [1,3,5,29,47–52,54]. By investigating the metabolic networks of one archaeon and one bacterium that each satisfy both their carbon and energy needs via $H_2$-dependent $CO_2$ reduction, we can identify their conserved common features. From comparative physiology [59] and from the standpoint of genes that trace to the LUCA [47,53], their shared features should reflect a state predating the divergence of the two prokaryotic domains. Subsets of the global prokaryotic $O_2$-independent network were obtained by parsing the genomes of the acetogen *Moorella thermoacetica* (Ace) and the methanogen *Methanococcus maripaludis* (Met). These were completed with reactions from corresponding manually curated genome-scale metabolic models [60,61], resulting in 1193 reactions for Ace and 920 for Met (electronic supplementary material, dataset S1B and S1C). Both the

Proc. R. Soc. B 287: 20192377

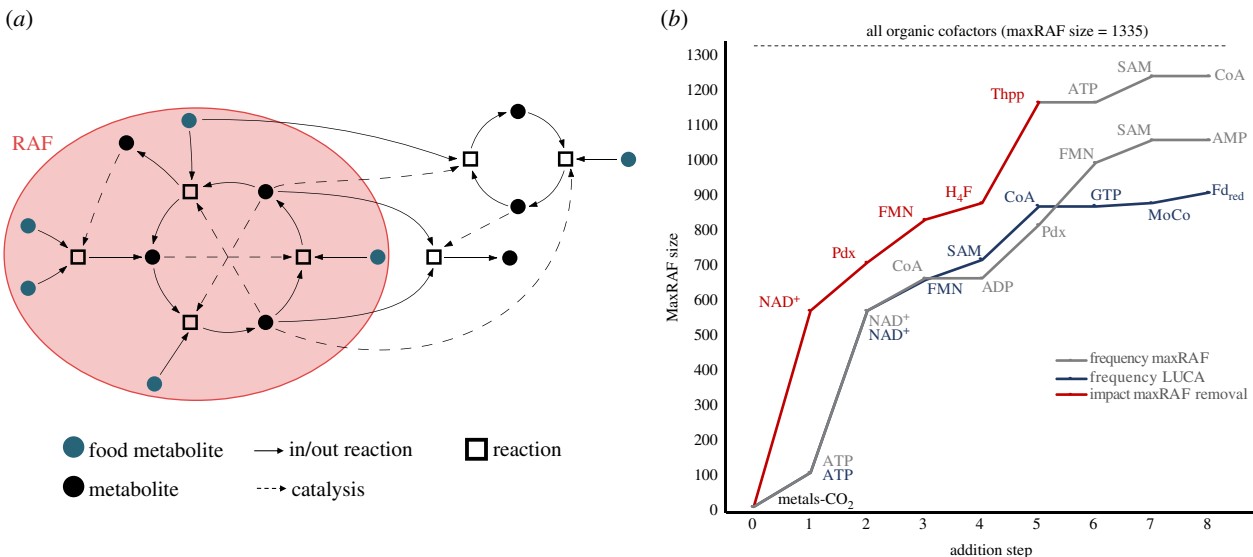

**Figure 2.** Autocatalysis in global metabolism expands with a small set of cofactors. (*a*) Schematic depiction of a reflexively autocatalytic food-generated network (RAF) highlighted (red ellipse) in a metabolic network. Food metabolites (green circles) may enter the RAF allowing subsequent reactions (squares) to occur and other metabolites (black circles) to be produced. Each reaction is catalysed by a metabolite in the network (catalysis shown in dashed arrows). (*b*) Increasing maxRAF sizes with the sequential addition to the initial food set (with only inorganic compounds) of the organic cofactors (i) with the highest impact on maxRAF size upon removal—NAD, Pdx, FMN H₄F, and Thpp (red)—followed by the three most frequent in the largest maxRAF with all organic cofactors added—ATP, SAM, and CoA (grey), (ii) most frequent in the maxRAF with all organic cofactors added (grey), and (iii) most frequent in enzymes predicted to be in LUCA [47] (blue) (Pdx, pyridoxal 5-phosphate; H₄F, tetrahydrofolate, Thpp, thiamine diphosphate, MoCo, molybdopterin; Fd$_{red}$, reduced ferredoxin). Top dashed line shows the maxRAF size obtained when all organic cofactors are added to the food set.

acetogen and the methanogen metabolic networks contain RAFs. When all organic cofactors are added to the food set, the maxRAFs contain 394 and 209 reactions for Ace and Met, respectively, spanning major KEGG functional categories (figure 3; electronic supplementary material, table S1 and figure S2 and S3 and dataset S2).

Carbon fixation and biosynthetic pathways are represented and amino acid biosynthesis is highly enriched in all maxRAFs, recovering autotrophic components of early autocatalytic metabolism. Note that, none of the food sets described so far in these analyses contained 'peptide'. Addition of the generic 'peptide' catalyst to the food set increases the maxRAF sizes obtained with the global anaerobic network, Met, and Ace by 93%, 47%, and 25%, respectively (electronic supplementary material, table S1). This indicates that adding protein catalysis expands cofactor-supported autocatalytic sets, but it does so to a much lesser degree in the metabolism of Met and Ace than it does in the global $O_2$-independent prokaryotic network.

## (d) Metabolism at the origin of LUCA was autocatalytic and autotrophic

The intersection of the Ace and Met maxRAFs should be more ancient than each of them individually. Three-quarters of the (smaller) Met maxRAF overlap with the (larger) Ace maxRAF in a connected network harbouring 172 reactions and 175 metabolites (figure 4; electronic supplementary material, figure S4; individual maxRAFs from Ace and Met in electronic supplementary material, figure S2 and S3). Six metabolites are disconnected, meaning the species interconvert them using different pathways; one example is that of glucose, catabolism of which arose after LUCA [62]. Highly connected food metabolites in the primordial network (more than 13 edges) include $H_2O$, ATP, protons, phosphate, $CO_2$, NAD⁺, pyruvate,

ammonia, diphosphate, coenzyme A, and AMP; highly connected non-food metabolites (more than eight edges) include ADP, NADH, and other pyridine dinucleotides, glyceraldehyde-3-phosphate, and acetyl-CoA (electronic supplementary material, dataset S3). The network is able to produce six amino acids—asparagine, aspartate, alanine, glycine, cysteine, and threonine—plus the two nucleosides UTP and CTP. Cytochromes and quinones do not figure into the network.

A different look at the primordial network reveals a hierarchical and highly connected organization (electronic supplementary material, figure S4) with a half-moon core structure with node degree varying from 49 to 4 (electronic supplementary material, dataset S3). Food molecules cluster in the most connected area, suggesting that autocatalytic metabolism is initiated by a handful of central substrate molecules with degree higher than 10.

## (e) Primordial metabolism is enriched in metal catalysis, ancient genes, and autotrophic functions

In search of the distinct contributions for autocatalysis, we tested for enrichment in individual catalysts, functions, and ancient genes encoding for reactions in the primordial network. There is a significant enrichment for metal and metal–sulfur cluster catalysis (figure 5*a*), whereas thiamine diphosphate (a carrier of C2 units in metabolism) is the only organic cofactor that is significantly enriched in catalysing the primordial network when compared with the global network, even though several others are present and essential for the network to grow (figure 5*b*). The primordial network is also enriched in reactions for amino acid biosynthesis, carbon metabolism, and 2-oxocarboxylic acid metabolism when compared with the global network (figure 5*b*). Comparing reactions in the primordial network to those catalysed by genes that can be traced to LUCA by

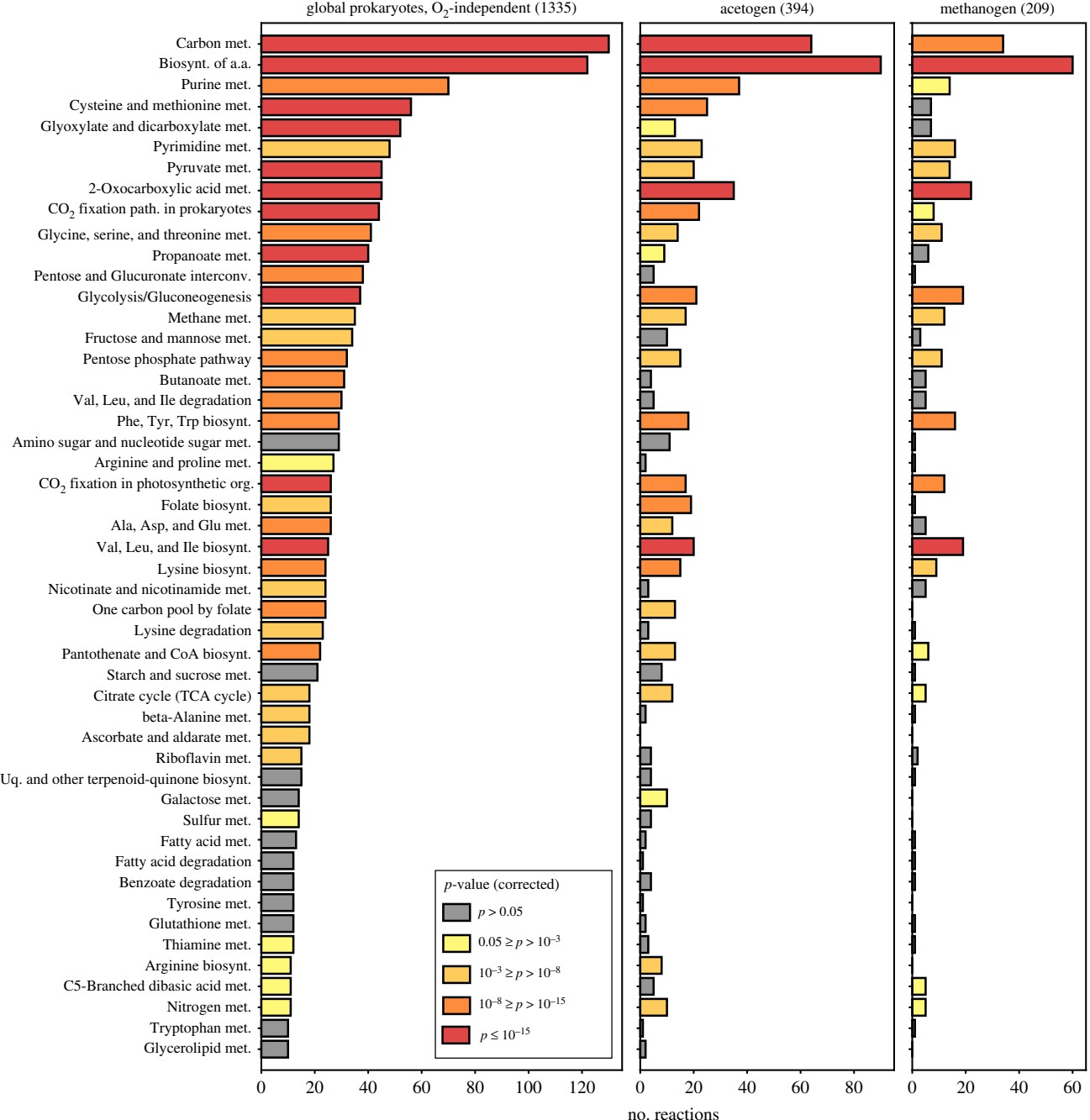

**Figure 3.** Autocatalytic networks point to an early autotrophic metabolism. The number of reactions in each functional category for three maxRAFs and functional enrichment relative to the global $O_2$-independent prokaryotic network. Colours represent bins of corrected $p$-values (Fisher's exact test with Benjamini–Hochberg false discovery rate (FDR) multiple-testing correction). From left to right, maxRAF obtained for (sizes in brackets): global $O_2$-independent prokaryotic network, acetogen (Ace) and methanogen (Met). Categories are sorted according to the number of reactions in the first maxRAF, from largest to smallest; only categories where this maxRAF had more than 10 reactions are shown.

independent phylogenetic criteria [47] uncovers highly significant enrichment relative to both the global network and its maxRAF (figure 5c). The maxRAF obtained within the primordial network contains 120 reactions and is enriched in amino acid and carbon metabolism but produces cysteine as the sole amino acid, which is noteworthy because cysteine is the hub of sulfur metabolism and also is the sole ligand for incorporating Fe–S and Fe–Ni–S clusters in proteins (electronic supplementary material, figure S5).

## (f) Autocatalysis, ATP, NAD, and monomers
Crucial catalysts can be identified by removing them from the food set. NAD$^+$ is strongly embedded in the RAF and its

removal reduces the size of the maxRAF by approximately 50% (electronic supplementary material, figure S6). The role of NADs, which we use to collectively designate NAD(P)$^+$ and NAD(P)H, in the food set of the largest maxRAFs is striking. It exhibits the strongest effect we observed for any cofactor. NAD donates and accepts hydride for redox-dependent reaction catalysis. The strict dependence of the largest maxRAFs upon NAD reflects the circumstance that microbial metabolism, without exception, always involves redox reactions [63]. Though Fe–S clusters are more ancient redox cofactors than NAD(P)H [64], they perform one-electron transfer reactions and have been replaced in evolution by NAD and other two-electron carriers [65,66]. Fe–S clusters also heavily impact maxRAF sizes, together with pyridoxal-5-phosphate and divalent metals.

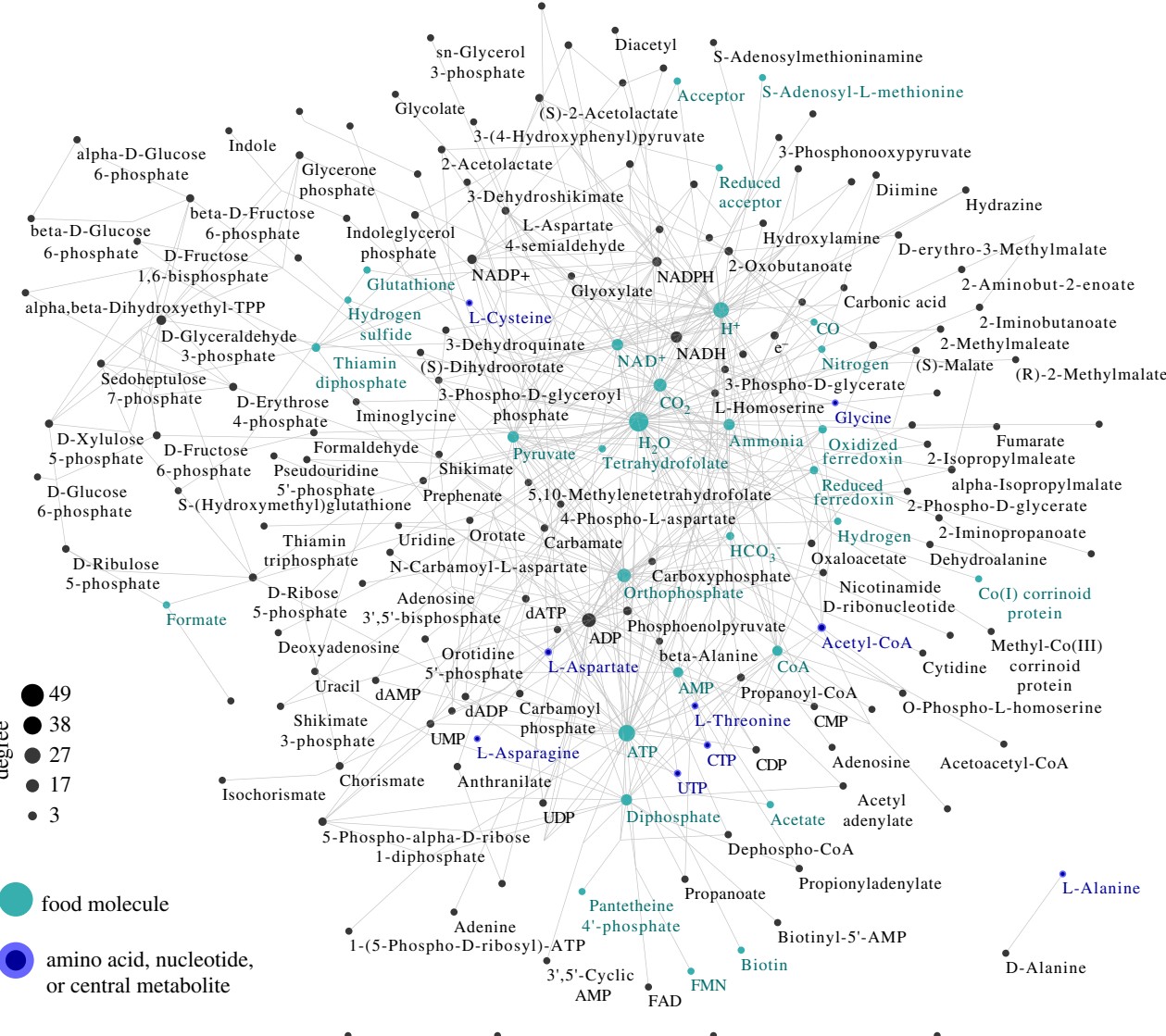

**Figure 4.** Core metabolism at the origin of the last universal common ancestor (LUCA). Intersection of the maxRAFs obtained with the networks of *Moorella thermoacetica* and *Methanococcus maripaludis* with a food set with organic cofactors (only metabolic interconversions are depicted; catalysis arcs are omitted for clarity). Six metabolites, including D-glucose and L-alanine (bottom), are in the intersection but disconnected from the remaining network. The node size is scaled according to the degree, with food molecules highlighted in green and relevant products in dark blue. 'Acceptor' and 'Reduced Acceptor' are abstract redox molecules as represented in KEGG metabolism.

Surprisingly, when we remove ATP from the food set of organic cofactors, this has no impact on the size of the maxRAF (the number of reactions in the maxRAF remains unchanged), both for the individual networks (electronic supplementary material, figure S6) and LUCA's network (electronic supplementary material, figure S4). Why does ATP removal from the food set with organic cofactors have no effect on RAFs? ATP is an essential intermediate in the maxRAF, but it is not required to kick-start it when other organics are present. This reflects the increasingly evident role of alternative energy currencies in primordial metabolism [6], such as acyl phosphates [29], thioesters [7], and reduced ferredoxin [49,67]. Alternative energy currencies are particularly common in anaerobes [49].

RAFs provided with a food set containing catalysts can generate amino acids and bases (figure 4), but the converse is not true: adding amino acids and bases to the simplest food set, which includes inorganic catalysts and $CO_2$ (electronic supplementary material, table S1), produces a miniscule 33-reaction maxRAF (electronic supplementary material, figure S7). The maxRAF contains 47 metabolites, 27 of which are food molecules. This indicates that primordial autocatalytic

networks embedded in microbial metabolism generated amino acids and bases using small-molecule catalysis.

## 3. Discussion

Autocatalytic networks are objects of molecular self-organization [8–12]. Their salient property in the study of early biochemical evolution is the capacity to grow in size and complexity. Compounds generated from the food set become part of the network, hence autocatalytic networks can start small and grow, in principle to a size approaching the complexity of metabolic networks of modern cells [15], and very little catalysis by individual elements is required for autocatalytic networks to emerge [19,20]. Reflexively autocatalytic and food-generated networks—RAFs—are a particularly interesting formalization of collectively autocatalytic sets, as they capture a property germane to life: they require a constant supply of an environmentally provided food source in order to grow [19]. In that sense, RAFs reflect metabolic networks in real cells, in that growth substrates are converted to end products, a proportion of which comprises the substance of cells.

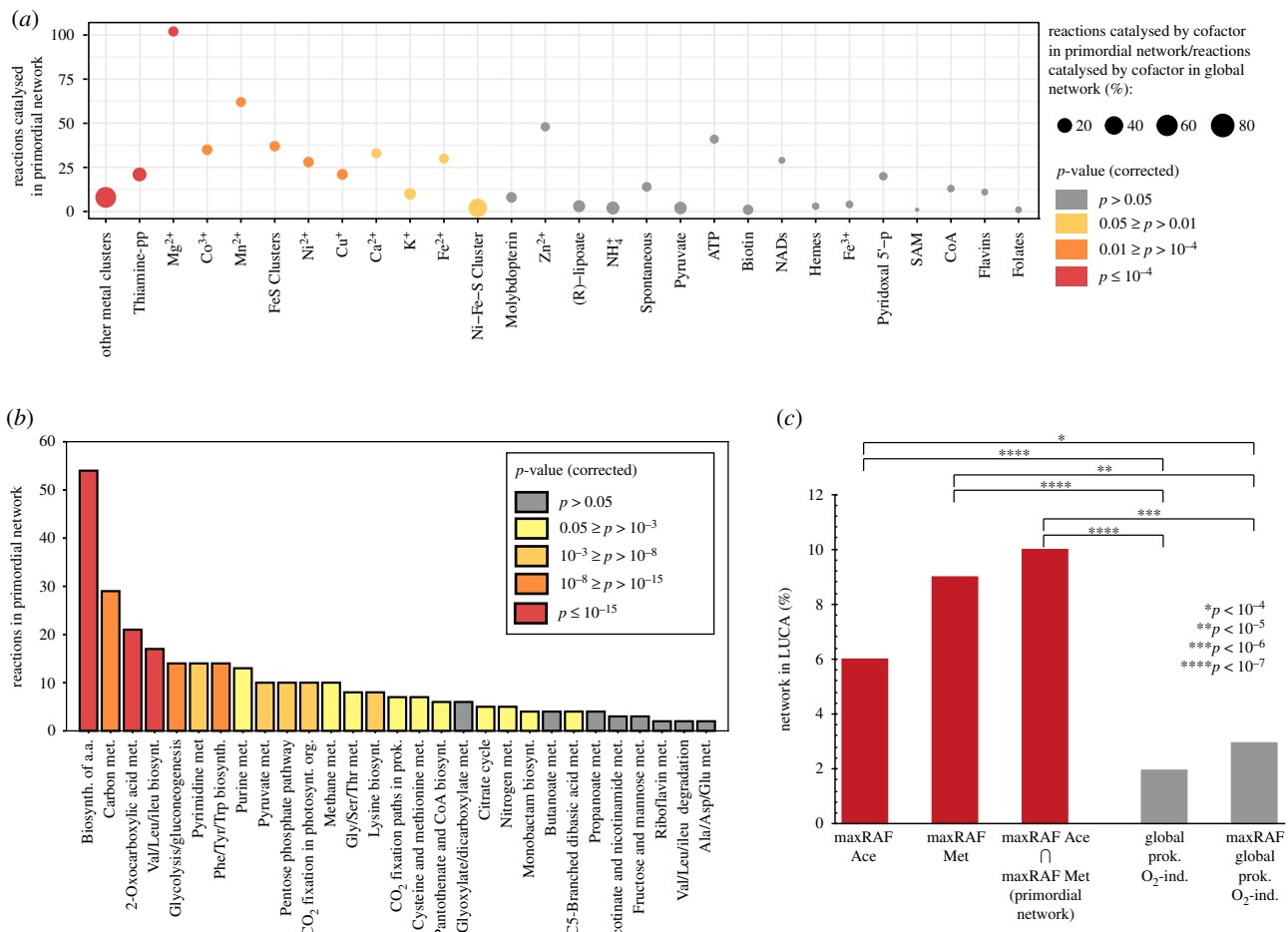

**Figure 5.** Properties of the core metabolism at the origin of the last universal common ancestor (LUCA). (a) The number of reactions catalysed by each cofactor in the primordial network (overlapping network between the acetogen and methanogen maxRAFs) and enrichment for each cofactor when compared with the number of reactions catalysed by that cofactor in the global $O_2$-independent prokaryotic network. Circle size indicates the ratio between the number of reactions in the primordial network and number of reactions in the global network catalysed by each cofactor; colour indicates the corrected $p$-value (Fisher's exact test with Benjamini–Hochberg FDR correction). (b) The number of reactions in each functional category in the primordial network and functional enrichment relative to the global $O_2$-independent prokaryotic network. Colour indicates bins of corrected $p$-values (Fisher's exact test with Benjamini–Hochberg FDR correction). (c) The proportion of metabolic networks predicted to be in LUCA [47] and enrichment of the individual maxRAFs and the primordial network (red) compared with the global network and the maxRAF obtained with it (grey) (Fisher's exact test).

But RAFs are far simpler than metabolism because they can start very small.

RAFs have not been applied in the study of the evolution of chemical networks that led to the metabolism of modern cells, themselves large natural autocatalytic networks. By embracing the simple and robust premise that reactions catalysed by simple molecules and inorganic compounds preceded metabolic reactions catalysed by enzymes [17,26,30], we have retooled RAFs into an analytical instrument to investigate the nature of metabolic evolution.

Our analyses started with the enzymatic and spontaneous reactions charted in modern metabolism and used RAFs as a filter to uncover elements with self-organizational properties, to address the nature of processes in the earliest phases of evolution, before the origin of eukaryotes and before the appearance of oxygen. We found evidence for a role of autocatalytic networks at the onset of metabolism. The largest RAF that we identified in the whole prokaryotic anaerobic biochemical space has 1335 reactions and points to early autotrophy. This RAF is larger than the genome size of the smallest free-living archaeon, *Methanothermus fervidus* [68]. With a genome coding for 1311 proteins and 50 RNA genes, *M. fervidus* lives from $H_2$ and $CO_2$ as carbon and energy sources (the food set) and requires only inorganic, geochemical nutrients, no other

cells for survival [69]. $H_2$ and $CO_2$ were present in abundance on the early Earth and may have given rise to the first metabolic pathways that brought forth the first archaeal and bacteria cells [3,6,47]. Our anaerobic RAF is, however, smaller than the reaction network in the smallest genome of bacteria that live from $H_2$ and $CO_2$, which is found in the acetogen *Thermoanaerobacter kivui*, encoding 2378 proteins [70].

*Methanothermus fervidus* and *T. kivui* harbour primitive forms of methanogenesis and acetogenesis in that they both lack cytochromes and quinones, suggesting that they represent energy metabolic relics from the earliest phases of biochemical evolution on the primordial Earth, before anaerobic respiratory chains had evolved [29]. To investigate this aspect further, we examined the best annotated metabolic networks existing for $H_2$-$CO_2$-dependent archaea and bacteria, the methanogen *Methanococcus maripaludis* and the acetogen *Moorella thermoacetica*. Remarkably, a food set containing only small abiogenic molecules and a handful of organic cofactors generates sizeable RAFs in each of the networks, with 209 and 394 reactions, respectively. The inclusion of organic molecules as catalysts in our food set is in line with a premise common to all scientific theories for the origin of life, namely that the environment provided starting material from which metabolism and life evolved. The small sizes of

maxRAFs compared to full metabolic networks in the genome-scale metabolic models are contingent upon two aspects. First, all reactions in the maxRAFs presented here require small-molecule catalysis. This excludes important edges in the network catalysed by peptides only, that allow for significant expansion, when the generic catalyst 'peptide' is added to the food set (electronic supplementary material, table S1). Moreover, in genome-scale models, a method known as gap-filling allows genome-scale metabolic models to be connected to the necessary degree for the production of all biomass components [71]. To avoid the introduction of noise and over-fitting of our model for autocatalysis, we have not performed gap-filling at any stage in this work. Note that the genome-scale models used here also require organic cofactors to produce biomass [60,61].

RAFs uncover elements of metabolic evolution that pre-date the divergence of archaea and bacteria from the LUCA. The intersection of the RAFs of *M. maripaludis* and *M. thermo-acetica* uncovers commonalities—a core, conserved network with 172 reactions that is enriched in metal catalysis and carbon-metal bonds [72] that points both to early autotrophy [5] and to the origin of the genetic repertoire of LUCA [47]. This conserved network does not produce all 20 amino acids, even though LUCA probably used all of them, given the universality of the genetic code. This apparent contradiction is reconciled by the virtual certainty that LUCA, before it became a free-living cell, was auxotrophic for some amino acids (and other components) that were provided by the environment [73]. Our model does not include catalysed transport reactions, and these would be an interesting addition in future formulations. Our results indicate that some enzyme catalysis had to be invented to allow for a sustained production of several amino acids. Our results also show that the kick-start of autocatalysis in anaerobic metabolism does not require ATP in the food set, even though ATP is an essential intermediate in all maxRAFs. This relates to the use of alternative energetic currencies in anaerobic prokaryotes [49] and recent findings that suggest that complexity in early metabolic reaction systems could have emerged without phosphate [6]. More importantly, NAD plays an essential role in kick-starting all sizeable maxRAFs obtained here. This underscores the special role of redox chemistry in primordial catalysis [49,74].

An important insight uncovered by RAFs is the observation that although a food set with organic cofactors sparks a large autocatalytic metabolic network that generates amino acids and bases, the opposite does not occur: adding amino acids and bases to the simplest food set (which includes inorganic catalysts and $CO_2$) only produces a minute RAF with 33 reactions. The result that autocatalytic networks detectable in microbial metabolism as RAFs generate amino acids and bases using small-molecule catalysts is in accord with the recent report of amino acid synthesis catalysed by native metals [35], and with the physiology of extant anaerobic auto-trophs: amino acids and bases are sequestered end-products of $H_2$- and $CO_2$-dependent metabolism, they are polymerised to make the substance of cells.

RAFs can serve as a guide for the identification and construction of larger, biologically relevant autocatalytic reaction networks. The synthesis of compounds characteristic of the metabolism of acetogens and methanogens, intermediates and end products of the acetyl-CoA pathway, and of the incomplete citric acid cycle from $CO_2$ using only the catalysis of native metals [3,32–36], as well as the demonstrated catalytic power of

organic cofactors without their enzymes including flavins [37], pyridoxal 5′-phosphate [38,39], SAM [40], NAD [41], and others [42–45] encourages the investigation of more complex autocatalytic networks in laboratory reactors. Individual reactions in the maxRAFs presented here can be tested in independent experiments with the catalysts assigned (electronic supplementary material, dataset S2). Though we have not explored subRAFs [75] embedded within the (larger) maxRAFs in the present work, this may be an interesting route for further investigation, amenable to laboratory experiments and/or the incorporation of experimental data [58]. The increasing availability of large-scale kinetic, thermodynamic, and inhibition data for metabolism will allow further exploration of RAFs [58]. The plausibility and sustainability of RAFs under various scenarios has been explored in extensive simulations [76].

Our results are directly relevant to two deeply divided schools of thought concerning the nature of chemical reactions at the origin of life: genetics first and metabolism first. The genetics first school, or RNA world, holds that the origin of RNA molecules marked the origin of life-like processes, and that RNA both self-replicated and possessed catalytic abilities that led to the emergence of biochemical reactions [77,78]. In that view, the origin of the bases that drove that process forward is decoupled from biochemical processes that are germane to modern cellular metabolism. Proponents of the RNA world [77,78] might criticize that we did not investigate the possible catalytic role of RNA in the metabolic networks presented here. However, no reactions in KEGG have RNA catalysts assigned in Uniprot, an observation concerning both the nature of catalysis in microbial metabolism and the limitations inherent to current databases. If we count RNA-related cofactors as representing the RNA world [26], the participation of RNA-like bases in RAFs is ubiquitous, but as monomers, with polymers playing no role. Hence, addition of a generic polymer 'RNA' has no impact on RAFs while the generic catalyst 'peptide' does, whereby we note that KEGG does not include the full processes of transcription, splicing, and translation, which require the genetic code, an innovation that arose subsequent to the phase of biochemical evolution probed here by RAFs.

The metabolism-first school holds that spontaneous (exergonic) chemical reactions preceded reactions catalysed by genetic material, and that those exergonic reactions continuously gave rise to substrate-product relationships [29,79]. From such reactions, more complex interaction networks with autocatalytic properties arose [16,80], in which elements of the set intervened in reactions of the set, providing structure and direction to product accumulation. Our results indicate that RNA monomers can readily arise from autocatalytic networks—though the converse is not true—and the nature of the products accumulated in RAFs will include nucleic acids. In other words, RAFs applied to ancient auto-trophic metabolism reveal a vector of autopoietic genesis that detects RNA as emergent from metabolism.

# 4. Material and methods

Detailed methods including annotation procedures and algorithms can be found in the electronic supplementary material.

Data accessibility. All biochemical data used here is publicly available in KEGG, Uniprot, and in the publications of the genome-scale metabolic models of *Moorella thermoacetica* [60] and *Methanococcus maripaludis* [61]. The curated and annotated networks are provided

in electronic supplementary material, dataset S1. A custom-made implementation of the maxRAF algorithm was used for the analysis in this paper and is available at https://www.canterbury.ac.nz/engineering/schools/mathematics-statistics/research/bio/downloads/raf/. An example of an input file (global prokaryotic O2-independent network, food set with all small molecules, abiotic carbon, and organic cofactors) is given in electronic supplementary material, dataset S4. A more general-purpose and interactive RAF analyser can be found online at https://github.com/husonlab/catlynet, including several more examples and explanations.

Authors' contributions. J.C.X. collected and integrated data from databases, curated models, and literature. J.C.X., M.S., S.K., and W.F.M. designed the project. W.H. and M.S. wrote the pseudocode and algorithm for detection of maxRAFs and W.H. performed maxRAF identification. J.C.X. and W.F.M. wrote the first manuscript draft. J.C.X., W.H., S.K., M.S., and W.F.M. contributed in data interpretation and writing the final manuscript.

Competing interests. We declare we have no competing interests.

Acknowledgements. We thank Filipa L. Sousa, Martina Preiner, and anonymous reviewers for comments.

Funding. This work was supported by grants from the European Research Council (666053) and the Volkswagen Foundation (93 046) to W.F.M. W.H. thanks the Institute for Advanced Study, University of Amsterdam, The Netherlands, for partial support in the form of a fellowship.

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
