## [Reviewer comments · Proceedings of the Royal Society B: Biological Sciences]

Review History

RSPB-2019-2377.R0 (Original submission)

Review form: Reviewer 1 (Eörs Szathmáry)

Recommendation

Accept with minor revision (please list in comments)

Scientific importance: Is the manuscript an original and important contribution to its field?

Excellent

General interest: Is the paper of sufficient general interest?

Good

Quality of the paper: Is the overall quality of the paper suitable?

Excellent

Is the length of the paper justified?

Yes

Should the paper be seen by a specialist statistical reviewer?

No

Do you have any concerns about statistical analyses in this paper? If so, please specify them explicitly in your report.

No

It is a condition of publication that authors make their supporting data, code and materials available - either as supplementary material or hosted in an external repository. Please rate, if applicable, the supporting data on the following criteria.

Is it accessible?

Yes

Is it clear?

Yes

Is it adequate?

Yes

Do you have any ethical concerns with this paper?

No

Comments to the Author

This is a clearly written, important contribution to the problem of the origin of metabolism, biochemical evolution, and life in general. Data and analysis are nicely presented.

There are some issues with embedding in broader literature, importance and implication of the results that warrant careful re-discussion.

My comments form a “network” conceptually, so I present them in a somewhat arbitrary order (natural language is sequential).

1. The authors stick to the RAF idea, where every reaction must be catalyzed by some member. This is a constraint that is likely to be too stringent. Let us agree that the more spontaneous reactions a proposed network has, the better for the origin of life. The authors mention “spontaneous” reactions, but referring to what is spontaneous TODAY. Question is what was spontaneous before (or even well before) LUCA. For an answer we need more chemical evolution experiments. In other words, a RAF in LUCA may well have been a CHEMICAL PALIMPSEST of the earliest autocatalytic systems.

2. Screening the recent (occasionally spectacular) experiments of the latter kind, one envisages a scenario from less to more RAF-like, but with autocatalysis conserved throughout – didactically speaking: from (networks similar to) the formose reaction to various RAFs presented in this work. Emphatically, we do not know much about autocatalysis in many recent chemical experiments, especially when a complex network unfolds. A completely recycling complex network can (actually, is likely) to contain autocatalytic subparts (more like formose than RAF).

3. There is an important distinction between facultative and obligate autocatalytic networks. In the former there is autocatalytic organization, but the system can be reconstructed by feedforward production. In the case latter you need at least one instantiation of the autocatalytic seed to kick start the system, but NO MORE. This means that the claim that a compound must be in the food seed can mean two things: either it must be “eaten” constantly, or it is needed only for “ignition” I have the impression the analysis is unclear on this important point. True? It does matter.

4. More generally, the authors are too parochial in their discussion of autocatalysis. The idea of small-molecule RAFs is conceptually important and likely historically relevant, but likely not exclusively so. In other words, different approaches to autocatalysis are not mutually exclusive in relevance, especially from an evolutionary point of view.

For revision I suggest to include some key references in line with the above points (also to improve scholarship):

Life and metabolism as autocatalytic organization, centred mainly on small molecules.

Obviously, Gánti (1971, and after). Note that this is the same “annus mirabilis” when Stu and

Eigen also produced their pioneering papers.

Coenzymes as forming auto-and heterocatalytic networks: King (1980). In my view he was exceptional (arguing with Eigen a lot – and being right most of the time).

Recycling systems and (non-RAF) autocatalytic networks: King (1982).

First empirical analysis of autocatalysis in the contemporary metabolic databases, facultative and obligate autocatalysis: Kun et al (2008)

Autocatalytic metabolic organization before enzymes and templates: obviously, Wachtershauser (several papers). Please note that his proposals were intermediate between non-RAF and RAF autocatalytic networks.

In sum, I think is a very valuable paper in need of some conceptual revision and possibly technical clarification. It will move the field forward.

Review form: Reviewer 2

Recommendation

Major revision is needed (please make suggestions in comments)

Scientific importance: Is the manuscript an original and important contribution to its field?

Excellent

General interest: Is the paper of sufficient general interest?

Good

Quality of the paper: Is the overall quality of the paper suitable?

Good

Is the length of the paper justified?

Yes

Should the paper be seen by a specialist statistical reviewer?

No

Do you have any concerns about statistical analyses in this paper? If so, please specify them explicitly in your report.

Yes

It is a condition of publication that authors make their supporting data, code and materials available - either as supplementary material or hosted in an external repository. Please rate, if applicable, the supporting data on the following criteria.

Is it accessible?

Yes

Is it clear?

Yes

Is it adequate?

Yes

Do you have any ethical concerns with this paper?

No

Comments to the Author

Xavier et al explored reflexively autocatalytic food-generated networks (RAFTs) in metabolism of potentially ancient microbes: presumed anaerobic prokaryotes, anaerobic autotrophs (a methanogen and an acetogen), and their intersection that is predicted to close to the last universal common ancestor (LUCA). Notably, they found that RAFTs are found in all the metabolic networks, and the larger RAFTs tend to form if inorganic molecules and cofactors are provided as food sources, when they assumed most of the reactions proceeds without enzyme but with cofactors and inorganic metals. A series of the analysis provides a fruitful information about the early evolution of metabolic networks in the origin of life (LUCA). The results themselves are interesting and new. However, before publication, I would like the authors to address the following comments, where I particularly questioned the plausibility of RAFTs described in the manuscript.

1. L41 & L74-81: The authors considered that RAFTs can be identified in modern metabolism by attributing the catalysis of enzymes to their metals and cofactors in prebiotic evolution, with evidence that several metals and cofactors can actually perform catalysis without enzymes. (1) Is there a sufficient basis that each metal and cofactor can generally catalyze the related enzymatic reactions in modern metabolism? (2) Can those reactions occur in realistic timescale to form RAFTs? (3) Although the authors raised several experimental evidence of metals- and cofactors-based catalysis of enzymatic reactions, are they true in other metals and cofactors investigated in the manuscript? If these assumptions were not sufficiently supported, although I found the presented results interesting, it is not very convincing to say that the RAFTs (of the analyzed sized) could form without enzymes and may have preceded the evolution of proteins.

2. L115: First of all, the authors should describe what 'peptide' means here (where the word appeared for the first time) instead of L185. In addition, I think the word choice is slightly misleading because peptides are generally considered small compounds of amino acids. In the manuscript, however, large protein enzymes that function without cofactors and metals, if any, are also attributed to 'peptides'.

3. L119: Did the authors mean 'six' Enzyme Commission instead of 'five'? The abbreviation of Enzyme Commission also varied throughout the manuscript (E.C. or EC).

4. Fig. 2, etc.: As shown in Fig. 2A, I suppose most RAFTs are not completely closed networks but they have a lot of side-reactions that produce by products. If there are too many side-reactions, RAFTs should not be stable. Furthermore, the authors did not consider stoichiometry in each reaction for the search of RAFTs. The efficiency of each reaction is also a black box. These facts made me question about the sustainability (or plausibility) of RAFTs, which the authors did not mention about. Although conceptually interesting, the meaningfulness of the analysis is therefore doubtful to me. Is it possible to give some estimates about the sustainability or plausibility of each of the analyzed RAFTs?

5. L154-156: I could not follow this description, because looking at the corresponding figure and legend, I read is as the addition of eight compounds with the greatest impact on removal.

6. Fig. 3 & 5: I could not follow what p-values represent. In other words, where does randomness come from?

7. Fig. 5A: Is there any reason why the authors used "Reactions in RAFT / reactions in network" as enrichment? It is slightly unusual to me because it does not consider the number of total reactions, and as a result, it is difficult to understand how enriched each cofactor-dependent reaction is. The authors may think about using "Ratio of reactions in RAFT / ratio of reactions in network", giving an indication of X-fold enrichment of each set of reactions.

8. Fig. 5B: Why did the authors show only the absolute increased reaction numbers? The extent of

enrichment should depend on the relative number of each reaction set in a network. Is it possible to show the extent of enrichment as in Fig. A or in my comment 7?

9. L224-227 & Fig. 5C: If I understood correctly, the authors found that 120 reactions of the reconstituted primordial network out of reactions that can be traced back to putative LUCA can form a (max) RAF. However, the primordial network contained a larger maxRAF of 172 reactions (cf. L196). Does this indicate that the gap of 52 reactions would represent some convergent evolution after that occurred after a diversification from LUCA to distinct species?

10. L245-247 & Fig. S6: (1) Where is the data of removing ATP? This lack made me wonder whether "no impact" means zero impact (even so, I think the authors should provide actual data in Fig. S6) or sufficiently small impact. (2) Does "spontaneous" in Fig. S6 mean removing all the spontaneous reactions simultaneously?

11. L370-376: I could not follow the authors' logic here. Addition of a generic RNA 'polymer' has no impact on RAFs, but I think that is simply because RNA catalysts are not associated with KEGG and Uniprot, the impact of RNA on RAFs may depend on what method (data source) to be used. A huge number of RNA catalysts are known, and I therefore believe RNAs, as well as peptides, may have had impact on RAFs in the early evolution of life. Moreover, at L72-74, the authors introduced a few examples of RAFs, but one of the works (17) is actually RNA-based.

Review form: Reviewer 3

Recommendation

Accept as is

Scientific importance: Is the manuscript an original and important contribution to its field?

Good

General interest: Is the paper of sufficient general interest?

Excellent

Quality of the paper: Is the overall quality of the paper suitable?

Good

Is the length of the paper justified?

Yes

Should the paper be seen by a specialist statistical reviewer?

Yes

Do you have any concerns about statistical analyses in this paper? If so, please specify them explicitly in your report.

No

It is a condition of publication that authors make their supporting data, code and materials available - either as supplementary material or hosted in an external repository. Please rate, if applicable, the supporting data on the following criteria.

Is it accessible?

Yes

Is it clear?

Yes

Is it adequate?

Yes

Do you have any ethical concerns with this paper?

No

Comments to the Author

The manuscript deals with an extremely important problem: identify Reflexively Autocatalytic Food-generated networks (RAF), which are self-sustaining networks that collectively catalyze all their reactions and might be the precursors of more complex molecular organizations. The existence of such objects would provide indirect evidence for a stepwise evolution leading to the origin of cells through processes of molecular self-organization.

The authors have analyzed various data sets to investigate different levels of ancient metabolism preserved in modern cells in order to identify RAF. Indeed they succeeded, to some extent, and the largest RAF that they have found comprises more than one thousand reactions in the whole prokaryotic anaerobic biochemical space.

The results of this manuscript are relevant for the various schools investigating the origin of life. I should add that I am not an expert of the kind of statistical analysis that the authors have performed.

Decision letter (RSPB-2019-2377.R0)

29-Jan-2020

Dear Dr Xavier:

Your manuscript has now been peer reviewed and the reviews have been assessed by an Associate Editor. The reviewers' comments (not including confidential comments to the Editor) and the comments from the Associate Editor are included at the end of this email for your reference. As you will see, the reviewers and the Editors have raised some concerns with your manuscript and we would like to invite you to revise your manuscript to address them.

Research ethics:

Use of animals and field studies:

Please submit a copy of your revised paper within three weeks. If we do not hear from you

within this time your manuscript will be rejected. If you are unable to meet this deadline please let us know as soon as possible, as we may be able to grant a short extension.

Best wishes,
 Professor Hans Heesterbeek
 mailto: proceedingsb@royalsociety.org

Associate Editor

Comments to Author:

We have now received three expert opinions on your manuscript. All three reviewers saw considerable value in the work. Reviewer 1, while generally positive, suggested that it was important to better contextualize your work in terms of the broader literature and notions of autocatalysis. Reviewer 2 was more skeptical about the plausibility of your hypothesis regarding the central role of RAFs in the evolution of metabolism and origin of life (eg, comment #1). This reviewer also pointed out areas where the results could potentially be more clearly presented, and the relevant statistics and definitions more clearly defined/explained.

Like the reviewers, I found the results to be intriguing. But, given the broad reach of this ms, I think it's important to address the issues raised by reviewer 2.

Reviewer(s)' Comments to Author:

Referee: 1

Comments to the Author(s)

This is a clearly written, important contribution to the problem of the origin of metabolism, biochemical evolution, and life in general. Data and analysis are nicely presented. There are some issues with embedding in broader literature, importance and implication of the results that warrant careful re-discussion.

My comments form a "network" conceptually, so I present them in a somewhat arbitrary order (natural language is sequential).

1. The authors stick to the RAF idea, where every reaction must be catalyzed by some member. This is a constraint that is likely to be too stringent. Let us agree that the more spontaneous reactions a proposed network has, the better for the origin of life. The authors mention "spontaneous" reactions, but referring to what is spontaneous TODAY. Question is what was spontaneous before (or even well before) LUCA. For an answer we need more chemical evolution experiments. In other words, a RAF in LUCA may well have been a CHEMICAL PALIMPSEST of the earliest autocatalytic systems.
2. Screening the recent (occasionally spectacular) experiments of the latter kind, one envisages a scenario from less to more RAF-like, but with autocatalysis conserved throughout – didactically speaking: from (networks similar to) the formose reaction to various RAFs presented in this work. Emphatically, we do not know much about autocatalysis in many recent chemical experiments, especially when a complex network unfolds. A completely recycling complex network can (actually, is likely) to contain autocatalytic subparts (more like formose than RAF).
3. There is an important distinction between facultative and obligate autocatalytic networks. In the former there is autocatalytic organization, but the system can be reconstructed by feedforward production. In the case latter you need at least one instantiation of the autocatalytic seed to kick start the system, but NO MORE. This means that the claim that a compound must be in the food seed can mean two things: either it must be "eaten" constantly, or it is needed only for "ignition" I have the impression the analysis is unclear on this important point. True? It does matter.

4. More generally, the authors are too parochial in their discussion of autocatalysis. The idea of small-molecule RAFs is conceptually important and likely historically relevant, but likely not exclusively so. In other words, different approaches to autocatalysis are not mutually exclusive in relevance, especially from an evolutionary point of view.

For revision I suggest to include some key references in line with the above points (also to improve scholarship):

Life and metabolism as autocatalytic organization, centred mainly on small molecules.

Obviously, Gánti (1971, and after). Note that this is the same “annus mirabilis” when Stu and Eigen also produced their pioneering papers.

Coenzymes as forming auto- and heterocatalytic networks: King (1980). In my view he was exceptional (arguing with Eigen a lot – and being right most of the time).

Recycling systems and (non-RAF) autocatalytic networks: King (1982).

First empirical analysis of autocatalysis in the contemporary metabolic databases, facultative and obligate autocatalysis: Kun et al (2008)

Autocatalytic metabolic organization before enzymes and templates: obviously, Wachtershauser (several papers). Please note that his proposals were intermediate between non-RAF and RAF autocatalytic networks.

In sum, I think is a very valuable paper in need of some conceptual revision and possibly technical clarification. It will move the field forward.

Referee: 2

Comments to the Author(s)

Xavier et al explored reflexively autocatalytic food-generated networks (RAFTs) in metabolism of potentially ancient microbes: presumed anaerobic prokaryotes, anaerobic autotrophs (a methanogen and an acetogen), and their intersection that is predicted to close to the last universal common ancestor (LUCA). Notably, they found that RAFTs are found in all the metabolic networks, and the larger RAFTs tend to form if inorganic molecules and cofactors are provided as food sources, when they assumed most of the reactions proceeds without enzyme but with cofactors and inorganic metals. A series of the analysis provides a fruitful information about the early evolution of metabolic networks in the origin of life (LUCA). The results themselves are interesting and new. However, before publication, I would like the authors to address the following comments, where I particularly questioned the plausibility of RAFTs described in the manuscript.

1. L41 & L74-81: The authors considered that RAFTs can be identified in modern metabolism by attributing the catalysis of enzymes to their metals and cofactors in prebiotic evolution, with evidence that several metals and cofactors can actually perform catalysis without enzymes. (1) Is there a sufficient basis that each metal and cofactor can generally catalyze the related enzymatic reactions in modern metabolism? (2) Can those reactions occur in realistic timescale to form RAFTs? (3) Although the authors raised several experimental evidence of metals- and cofactors-based catalysis of enzymatic reactions, are they true in other metals and cofactors investigated in the manuscript? If these assumptions were not sufficiently supported, although I found the presented results interesting, it is not very convincing to say that the RAFTs (of the analyzed sized) could form without enzymes and may have preceded the evolution of proteins.

2. L115: First of all, the authors should describe what ‘peptide’ means here (where the word appeared for the first time) instead of L185. In addition, I think the word choice is slightly misleading because peptides are generally considered small compounds of amino acids. In the manuscript, however, large protein enzymes that function without cofactors and metals, if any, are also attributed to ‘peptides’.

3. L119: Did the authors mean ‘six’ Enzyme Commission instead of ‘five’? The abbreviation of Enzyme Commission also varied throughout the manuscript (E.C. or EC).

4. Fig. 2, etc.: As shown in Fig. 2A, I suppose most RAFs are not completely closed networks but they have a lot of side-reactions that produce by products. If there are too many side-reactions, RAFs should not be stable. Furthermore, the authors did not consider stoichiometry in each reaction for the search of RAFs. The efficiency of each reaction is also a black box. These facts made me question about the sustainability (or plausibility) of RAFs, which the authors did not mention about. Although conceptually interesting, the meaningfulness of the analysis is therefore doubtful to me. Is it possible to give some estimates about the sustainability or plausibility of each of the analyzed RAFs?

5. L154-156: I could not follow this description, because looking at the corresponding figure and legend, I read is as the addition of eight compounds with the greatest impact on removal.

6. Fig. 3 & 5: I could not follow what p-values represent. In other words, where does randomness come from?

7. Fig. 5A: Is there any reason why the authors used "Reactions in RAF / reactions in network" as enrichment? It is slightly unusual to me because it does not consider the number of total reactions, and as a result, it is difficult to understand how enriched each cofactor-dependent reaction is. The authors may think about using "Ratio of reactions in RAF / ratio of reactions in network", giving an indication of X-fold enrichment of each set of reactions.

8. Fig. 5B: Why did the authors show only the absolute increased reaction numbers? The extent of enrichment should depend on the relative number of each reaction set in a network. Is it possible to show the extent of enrichment as in Fig. A or in my comment 7?

9. L224-227 & Fig. 5C: If I understood correctly, the authors found that 120 reactions of the reconstituted primordial network out of reactions that can be traced back to putative LUCA can form a (max) RAF. However, the primordial network contained a larger maxRAF of 172 reactions (cf. L196). Does this indicate that the gap of 52 reactions would represent some convergent evolution after that occurred after a diversification from LUCA to distinct species?

10. L245-247 & Fig. S6: (1) Where is the data of removing ATP? This lack made me wonder whether "no impact" means zero impact (even so, I think the authors should provide actual data in Fig. S6) or sufficiently small impact. (2) Does "spontaneous" in Fig. S6 mean removing all the spontaneous reactions simultaneously?

11. L370-376: I could not follow the authors' logic here. Addition of a generic RNA 'polymer' has no impact on RAFs, but I think that is simply because RNA catalysts are not associated with KEGG and Uniprot, the impact of RNA on RAFs may depend on what method (data source) to be used. A huge number of RNA catalysts are known, and I therefore believe RNAs, as well as peptides, may have had impact on RAFs in the early evolution of life. Moreover, at L72-74, the authors introduced a few examples of RAFs, but one of the works (17) is actually RNA-based.

Referee: 3

Comments to the Author(s)

The manuscript deals with an extremely important problem: identify Reflexively Autocatalytic Food-generated networks (RAF), which are self-sustaining networks that collectively catalyze all their reactions and might be the precursors of more complex molecular organizations. The existence of such objects would provide indirect evidence for a stepwise evolution leading to the origin of cells through processes of molecular self-organization.

The authors have analyzed various data sets to investigate different levels of ancient metabolism preserved in modern cells in order to identify RAF. Indeed they succeeded, to some extent, and

the largest RAF that they have found comprises more than one thousand reactions in the whole prokaryotic anaerobic biochemical space. The results of this manuscript are relevant for the various schools investigating the origin of life. I should add that I am not an expert of the kind of statistical analysis that the authors have performed.

Author's Response to Decision Letter for (RSPB-2019-2377.R0)

See Appendix A.

Decision letter (RSPB-2019-2377.R1)

11-Feb-2020

Dear Dr Xavier

I am pleased to inform you that your manuscript RSPB-2019-2377.R1 entitled "Autocatalytic chemical networks at the origin of metabolism" has been accepted for publication in Proceedings B.

The Associate Editor has recommended publication, but also suggests some (very) minor revisions to your manuscript. Therefore, I invite you to respond to the comments and revise your manuscript. Because the schedule for publication is very tight, it is a condition of publication that you submit the revised version of your manuscript within 7 days. If you do not think you will be able to meet this date please let us know.

- 1) A text file of the manuscript (doc, txt, rtf or tex), including the references, tables (including captions) and figure captions. Please remove any tracked changes from the text before submission. PDF files are not an accepted format for the "Main Document".
- 2) A separate electronic file of each figure (tiff, EPS or print-quality PDF preferred). The format should be produced directly from original creation package, or original software format. PowerPoint files are not accepted.

3) Electronic supplementary material: this should be contained in a separate file and where possible, all ESM should be combined into a single file. All supplementary materials accompanying an accepted article will be treated as in their final form. They will be published alongside the paper on the journal website and posted on the online figshare repository. Files on figshare will be made available approximately one week before the accompanying article so that the supplementary material can be attributed a unique DOI.

Sincerely,

Professor Hans Heesterbeek

Associate Editor:

Board Member

Comments to Author:

The authors have responded to all reviewer comments/concerns, and edited the ms accordingly. Below are just a few minor suggestions for clarity.

Lines 70-72: sentence is unclear. Is there a typo?

Lines 74-76: Sentence is a bit difficult to read.

Lines 203-205: Info. appears repetitive given info added in lines 128-130.

lines 439-442: Language in parentheses makes sentence overly dense.

Author's Response to Decision Letter for (RSPB-2019-2377.R1)

See Appendix B.

Decision letter (RSPB-2019-2377.R2)

12-Feb-2020

Dear Dr Xavier

I am pleased to inform you that your manuscript entitled "Autocatalytic chemical networks at the origin of metabolism" has been accepted for publication in Proceedings B.

Open Access

Paper charges

Sincerely,
Proceedings B
mailto: proceedingsb@royalsociety.org

Appendix A

Associate Editor

Comments to Author:

We have now received three expert opinions on your manuscript. All three reviewers saw considerable value in the work. Reviewer 1, while generally positive, suggested that it was important to better contextualize your work in terms of the broader literature and notions of autocatalysis. Reviewer 2 was more skeptical about the plausibility of your hypothesis regarding the central role of RAFs in the evolution of metabolism and origin of life (eg, comment #1). This reviewer also pointed out areas where the results could potentially be more clearly presented, and the relevant statistics and definitions more clearly defined/explained.

Like the reviewers, I found the results to be intriguing. But, given the broad reach of this ms, I think it's important to address the issues raised by reviewer 2.

Xavier et al.: We thank the editor for the thoughtful comments on our paper and his or her expression of interest in its content. We acknowledge that in the original submission some aspects were not clearly presented for a general readership and some important literature was not included, and we have gone to great effort to resolve this. Please find below our replies (in black and cursive font) to the referees' comments (in grey). Please note that line numbers mentioned in the reply correspond to those in the tracked-changed version of the manuscript, appended below.

Reviewer(s)' Comments to Author:

Referee: 1

Comments to the Author(s)

This is a clearly written, important contribution to the problem of the origin of metabolism, biochemical evolution, and life in general. Data and analysis are nicely presented.

There are some issues with embedding in broader literature, importance and implication of the results that warrant careful re-discussion.

My comments form a “network” conceptually, so I present them in a somewhat arbitrary order (natural language is sequential).

1. The authors stick to the RAF idea, where every reaction must be catalyzed by some

member. This is a constraint that is likely to be to stringent. Let us agree that the more spontaneous reactions a proposed network has, the better for the origin of life. The authors mention “spontaneous” reactions, but referring to what is spontaneous TODAY. Question is what was spontaneous before (or even well before) LUCA. For an answer we need more chemical evolution experiments. In other words, a RAF in LUCA may well have been a CHEMICAL PALIMPSEST of the earliest autocatalytic systems.

Xavier et al.: This is a very important point that R1 raises, which we are glad to clarify both here and in the manuscript (lines 70-76). The RAF formulation indeed asks all reactions to be catalyzed at some point. But, importantly, not from the ignition of the network. Another type of autocatalytic network (CAFs) imposes that restriction: that all reactions being catalyzed by their assigned cofactors must be so since the ignition of the network. But given that the RAF formulation relaxes this requirement, the RAFs we show do portray the transition R1 points to: we ask the network to be initiated, slowly, and for its catalysts to be produced at some point. Moreover, there may exist many different routes by which a given RAF can form, depending on the order of those initially spontaneous reactions, and more efficient cofactors that are also assigned as possible catalysts for the same reaction can be produced later and take over catalysis, increasing speed and efficiency. So, RAFs do reflect what R1 points to – the emergence of more efficient autocatalytic networks that become palimpsests of the original ones.

2. Screening the recent (occasionally spectacular) experiments of the latter kind, one envisages a scenario from less to more RAF-like, but with autocatalysis conserved throughout – didactically speaking: from (networks similar to) the formose reaction to various RAFs presented in this work. Emphatically, we do not know much about autocatalysis in many recent chemical experiments, especially when a complex network unfolds. A completely recycling complex network can (actually, is likely) to contain autocatalytic subparts (more like formose than RAF).

Xavier et al.: This is indeed the case. A maxRAF may contain several sub-RAFTs, a subject explored in (Hordijk, W., Steel, M. and Kauffman, S. (2012). The Structure of Autocatalytic Sets: Evolvability, Enablement, and Emergence. Acta Biotheoretica 60:379-392). It was outside of the scope of this paper to find and analyze subRAFTs, but we agree with R1 that it

will be a very interesting development to explore in upcoming work, and we now mention this in the Discussion (lines 384-386).

3. There is an important distinction between facultative and obligate autocatalytic networks. In the former there is autocatalytic organization, but the system can be reconstructed by feedforward production. In the case latter you need at least one instantiation of the autocatalytic seed to kick start the system, but NO MORE. This means that the claim that a compound must be in the food sed can mean two things: either it must be “eaten” constantly, or it is need only for “ignition” I have the impression the analysis is unclear on this important point. True? It does matter.

Xavier et al.: We have to be careful here on terminology. At least three of us have a very long record of work on networks, much of it mathematical where definitions are key. We do not use the terms facultative or obligate, though what the referee describes here is similar to differences between RAFs and CAFs. As mentioned in the reply above, RAFs can work without a catalyst Y at reaction X to start and produce the catalyst Y for X (and other reactions) and accelerate, corresponding to a facultative requirement, if one will. In CAFs the catalyst Y can be produced, but if reaction X requires Y before Y is produced (vector, directed) then X does not take place, unless Y is in the food set, corresponding to an obligate requirement. Both RAFs and CAFs can contain spontaneous reactions (no catalyst at all). Summa sumarum, it does matter and the distinction between RAFs and CAFs is now explained with a brief new passage (lines 70-76). For biologists, the terms facultative and obligate relate to the elements of the food set (no food no life) and do not apply to spontaneous reactions, not so for networks. Also take the case of metals, always food, never a product, and essential catalysts for anything halfway biological. The analysis is very specific on this point (we only report RAFs here), the text is now too, many thanks!

4. More generally, the authors are too parochial in their discussion of autocatalysis. The idea of small-molecule RAFs is conceptually important and likely historically relevant, but likely not exclusively so. In other words, different approaches to autocatalysis are not mutually exclusive in relevance, especially from an evolutionary point of view.

For revision I suggest to include some key references in line with the above points (also to improve scholarship):

Life and metabolism as autocatalytic organization, centred mainly on small molecules. Obviously, Gánti (1971, and after). Note that this is the same “annus mirabilis” when Stu and Eigen also produced their pioneering papers.

Coenzymes as forming auto- and heterocatalytic networks: King (1980). In my view he was exceptional (arguing with Eigen a lot—and being right most of the time).

Recycling systems and (non-RAF) autocatalytic networks: King (1982).

First empirical analysis of autocatalysis in the contemporary metabolic databases, facultative and obligate autocatalysis: Kun et al (2008)

Autocatalytic metabolic organization before enzymes and templates: obviously, Wachtershauser (several papers). Please note that his proposals were intermediate between non-RAF and RAF autocatalytic networks.

Xavier et al.: We agree with R1 that the literature (s)he points to is indeed relevant and was missing in the initial submission, and are grateful for the suggestions. We now cite Gánti (1971), Eigen (1971) and King (1978) next to the previously cited Kauffman (1971) in two locations (line 55 and line 286). We also now cite Wachtershäuser (1990), cite and refer to the work by Kun et al. (2008) (line 59-60).

In sum, I think is a very valuable paper in need of some conceptual revision and possibly technical clarification. It will move the field forward.

Xavier et al.: We thank R1 for such gratifying words, and for her or his important notes regarding technical clarity and conceptual contextualization in the literature. We believe that after addressing those during revision our paper has improved considerably.

Referee: 2

Comments to the Author(s)

Xavier et al explored reflexively autocatalytic food-generated networks (RAFTs) in metabolism of potentially ancient microbes: presumed anaerobic prokaryotes, anaerobic autotrophs (a methanogen and an acetogen), and their intersection that is predicted to close to the last universal common ancestor (LUCA). Notably, they found that RAFTs are found in all the metabolic networks, and the larger RAFTs tend to form if inorganic molecules and cofactors are provided as food sources, when they assumed most of the reactions proceeds without

enzyme but with cofactors and inorganic metals. A series of the analysis provides a fruitful information about the early evolution of metabolic networks in the origin of life (LUCA). The results themselves are interesting and new.

Xavier et al.: We thank R2 for the kind words and the interest in our manuscript.

However, before publication, I would like the authors to address the following comments, where I particularly questioned the plausibility of RAFs described in the manuscript.

1. L41 & L74-81: The authors considered that RAFs can be identified in modern metabolism by attributing the catalysis of enzymes to their metals and cofactors in prebiotic evolution, with evidence that several metals and cofactors can actually perform catalysis without enzymes. (1) Is there a sufficient basis that each metal and cofactor can generally catalyze the related enzymatic reactions in modern metabolism? (2) Can those reactions occur in realistic timescale to form RAFs? (3) Although the authors raised several experimental evidence of metals- and cofactors-based catalysis of enzymatic reactions, are they true in other metals and cofactors investigated in the manuscript? If these assumptions were not sufficiently supported, although I found the presented results interesting, it is not very convincing to say that the RAFs (of the analyzed sized) could form without enzymes and may have preceded the evolution of proteins.

Xavier et al.: We understand R2's concern, and we now add literature that supports our assumption, particularly: supporting the conceptual assumption—Penny (2005), Shapiro (2006) and Keller et al. (2015); supporting the experimentally-demonstrated, non-enzymatic activity of further organic cofactors than previously mentioned—Coenzyme A, thiamine diphosphate and folates (now covering all cofactors with significant impact in the maxRAF size (Fig. S6)); adding further experimental evidence for non-enzymatic metal catalysis (lines 88-91).

However, the largest maxRAF we identify consists of 1335 reactions annotated with 48 catalysts or pools thereof. It is therefore intractable to search for the experimental feasibility of all reactions with all cofactors and metals in the literature, for this reason we provided a number of particular experimental examples, previously seven, now fifteen (lines 88-91). If we were to provide one reference per cofactor-catalyzed reaction in the manuscript, it would take

over the full bibliography, and we would have a (much-required!) review of non-enzymatic catalysis. Moreover, given that absence of proof is not proof of absence, if there is no experimental evidence for the activity of one of these cofactors without the enzyme today, that does not imply that it does not occur. One of our expectations is to inspire experimental work that will test the catalysis we point to as central for RAF formation. As stressed by Gánti in his Principles of Life, p.136-7 “In principle, every reaction can take place without enzymes, since enzymes merely accelerate reactions”. In the case of slow reactions, simpler cofactors and metals—and temperature—have the greatest potential to solve the problem (Stockbridge et al. 2010).

2. L115: First of all, the authors should describe what ‘peptide’ means here (where the word appeared for the first time) instead of L185. In addition, I think the word choice is slightly misleading because peptides are generally considered small compounds of amino acids. In the manuscript, however, large protein enzymes that function without cofactors and metals, if any, are also attributed to ‘peptides’.

Xavier et al.: We thank R2 for pointing out this oversight from our side, we now explain the operational catalyst “peptide” the first time it is mentioned (lines 128-130). Regarding the word choice, we believe that at the origin of the first self-sustaining networks, the genetic code was yet to be invented. The words “protein” and “enzyme” imply the complexity of the genetic code, transcription and translation. They also can connote “peptide plus cofactor(s) and/or metals and/or prosthetic groups”. We therefore chose the word peptide, as the simplest unambiguous term, to refer to potential non-cofactor catalysis performed by a (potentially small) non-encoded molecule assembled via peptide bonds at the advent of biochemistry. We do not specify ribosomes, enzymatic but non-ribosomal peptide synthesis, non-enzymatic but catalyzed peptide synthesis, or noncatalyzed peptide synthesis. Peptide refers to all. We have given this considerable thought and discussion, we find peptide to be the best term for our purpose. We beg R2’s indulgence here, please.

3. L119: Did the authors mean ‘six’ Enzyme Commission instead of ‘five’? The abbreviation of Enzyme Commission also varied throughout the manuscript (E.C. or EC).

Xavier et al.: We are extremely grateful that R2 found this oversight, it should indeed have read “six”, which we have now corrected (line 134). We have also corrected the Enzyme

Commission abbreviation to read “EC” in the locations where it previously read “E.C.” (line 134, Figure 1).

4. Fig. 2, etc.: As shown in Fig. 2A, I suppose most RAFs are not completely closed networks but they have a lot of side-reactions that produce by products. If there are too many side-reactions, RAFs should not be stable. Furthermore, the authors did not consider stoichiometry in each reaction for the search of RAFs. The efficiency of each reaction is also a black box. These facts made me question about the sustainability (or plausibility) of RAFs, which the authors did not mention about. Although conceptually interesting, the meaningfulness of the analysis is therefore doubtful to me. Is it possible to give some estimates about the sustainability or plausibility of each of the analyzed RAFs?

Xavier et al.: Since the food set is assumed to be freely available, and a RAF is F-generated, it has been formally shown that RAFs satisfy the condition that they are stoichiometrically sustainable (this is established in Lemma 4.1(i) of Steel, M., Hordijk, W., and Smith, J. (2013). Minimal autocatalytic networks. Journal of Theoretical Biology 332: 96-107). This relates to the correspondence between RAFs and chemical organizations (which are stoichiometrically sustainable by definition) and has been demonstrated in Theorem 2(ii) of (Hordijk, W., Steel, M. and Dittrich, P. (2018). Autocatalytic sets and chemical organizations: Modeling self-sustaining reaction networks at the origin of life. New Journal of Physics 20:015011). To summarize, yes, they are sustainable as long as the food set never becomes limiting, a theoretical/mathematical condition. Note that feedback inhibition or product inhibition (for example as in thermodynamics) is not yet an element of implementable RAF models at this scale. We hope that there will be more curated data regarding inhibition, thermodynamics and kinetics for metabolic reactions (in the large numbers we analyze here, genome-scale) to explore this further in the future. Three of us have expressed this interest in a recent publication (Steel M, Hordijk W and Xavier JC, J R Soc Interface 2019:16). However, we note that extensive dynamical simulations have been performed to show the plausibility and sustainability of RAF sets under various scenarios (Serra and Villani. Protocells. Springer, 2017). We now mention this in the Discussion (lines 386-389).

5. L154-156: I could not follow this description, because looking at the corresponding figure and legend, I read is as the addition of eight compounds with the greatest impact on removal.

Xavier et al.: We thank R2 for pointing out this lack of clarity. Only five organic cofactors (NAD, Pdx, FMN H4F and Thpp – Figure S6) had significant impact on maxRAF size upon individual removal from the food set, so we added the three next most-frequent cofactors in that simulation to try and achieve a larger maxRAF (ATP, SAM and CoA). We now update the figure and explain this clearly in the figure caption (lines 474-477).

6. Fig. 3 & 5: I could not follow what p-values represent. In other words, where does randomness come from?

Xavier et al.: A contingency table was built for each statistical test, for example, regarding Figure 3:

	Reactions in RAF	Reactions not in RAF
Reactions in Category	a	b
Reactions not in Category	c	d

Therefore

$(a+b+c+d) = \text{global network}$

$(a+c) = \text{RAF}$

The p-value refers to the probability of having at least as many reactions as seen in a category in a RAF (a) if we were to select a random pick of reactions the same size of the RAF (a+c) from the global network (a+b+c+d). For this we used a one-tailed Fisher test (and with a Benjamini/Hochberg correction as we are testing for all categories).

The procedure was exactly the same for all other tests. We now expand on this in the Supplementary Methods for clarity.

7. Fig. 5A: Is there any reason why the authors used “Reactions in RAF / reactions in network” as enrichment? It is slightly unusual to me because it does not consider the number of total reactions, and as a result, it is difficult to understand how enriched each cofactor-dependent reaction is. The authors may think about using “Ratio of reactions in RAF / ratio of reactions in network”, giving an indication of X-fold enrichment of each set of reactions.

Xavier et al.: We believe we have not explained this detail sufficiently and clearly in the figure and legend and there is a misunderstanding. "Reactions in network" is the total reactions catalyzed by X cofactor in the larger, global network. The ratio (circle size) is also not used "as enrichment"; a statistical test, as previously described, was used to test for enrichment for each cofactor, and enrichment is shown in the circle colors. Moreover, we do not follow what R2 means with "how enriched each cofactor-dependent reaction is", as one cofactor-dependent reaction cannot be enriched, it is always catalyzed by the same cofactor(s). A network (in this case, the primordial network) can be enriched in reactions catalyzed by X cofactor, when compared to the amount of reactions catalyzed by same X cofactor in the global network, were we to select a random network of the same size (statistical test described above). We now change the figure to read "Reactions catalyzed by cofactor in primordial network/reactions catalyzed by Cofactor in Global network" for circle sizes, and the y axis to "Reactions catalyzed in primordial network". We also revised the figure legend accordingly (lines 502-509) and amended the explanation of the statistical tests (as described above) in the Supplementary Methods to help clarify this issue.

8. Fig. 5B: Why did the authors show only the absolute increased reaction numbers? The extent of enrichment should depend on the relative number of each reaction set in a network. Is it possible to show the extent of enrichment as in Fig. A or in my comment 7?

Xavier et al.: This seems to be another misunderstanding, as the bars do not show "increased reaction numbers", but the total of reactions in each category in the primordial network (as in Fig. 3, which shows the total reactions in each category in each maxRAF), colored according to enrichment when compared to the largest network after a Fisher one tailed test (explained above). So the enrichment, as R2 says, depends on the relative number of each set in the network, as described above. We now change the y axis label to "Reactions in primordial network" and amended the figure legend accordingly (lines 508-509).

9. L224-227 & Fig. 5C: If I understood correctly, the authors found that 120 reactions of the reconstituted primordial network out of reactions that can be traced back to putative LUCA can form a (max) RAF. However, the primordial network contained a larger maxRAF of 172 reactions (cf. L196). Does this indicate that the gap of 52 reactions would represent

some convergent evolution after that occurred after a diversification from LUCA to distinct species?

Xavier et al.: This is interesting. Indeed, the maxRAF obtained with the primordial network with 120 reactions is smaller than the primordial network itself, which has 172 reactions but is not a maxRAF. Why is it not a maxRAF? The 172-reaction intersection contains some nodes and some edges that are present in the two prokaryotic maxRAFs—but—the nature of connectivity among these 172 reactions is such that only a subset thereof generates a maxRAF (120 reactions). That is, portions of the acetogen and methanogen maxRAFs that produced molecules that held those 52 reactions in the respective individual maxRAFs (which were much larger) were not in the intersection – note that the food set was the same. Hence the loss of reactions. We were careful in our wording that the 172-reaction network contains but is not itself a maxRAF. We are grateful for R2 raising this point, we adjusted the text accordingly (there was one occurrence of the word autocatalytic that may have caused the problem – line 346, now corrected). With regard to evolutionary vectors, this would more likely correspond to divergence from a common ancestor in our view than convergence, but we have no way to test that. It could also relate to differentiation of food sets and selection/adaptation for new reactions. Many thanks to R2 for picking up this point!

10. L245-247 & Fig. S6: (1) Where is the data of removing ATP? This lack made me wonder whether “no impact” means zero impact (even so, I think the authors should provide actual data in Fig. S6) or sufficiently small impact. (2) Does "spontaneous" in Fig. S6 mean removing all the spontaneous reactions simultaneously?

Xavier et al.: With no impact we mean exactly zero impact—no reactions are lost in the maxRAF if ATP is removed from the food set. We now clarify this in the main text (line 268), and mention in the caption of Figure S6 that compounds with zero impact, as ATP, are not shown (the x axis would become too large with compounds with no bars if we were to include all in the plot).

Regarding the removal of spontaneous reactions, they are indeed indirectly removed. These 147 reactions are assigned an operational catalyst “spontaneous” that is normally always added in the food set (see Supplementary Methods and Table S1). In this simulation we remove “spontaneous” from the food set, as done with other cofactors, but keep the reactions

in the input network. However, as these reactions are not assigned any other catalyst, they will necessarily never be part of the maxRAF, because “spontaneous” cannot be produced by the network.

11. L370-376: I could not follow the authors’ logic here. Addition of a generic RNA ‘polymer’ has no impact on RAFs, but I think that is simply because RNA catalysts are not associated with KEGG and Uniprot, the impact of RNA on RAFs may depend on what method (data source) to be used. A huge number of RNA catalysts are known, and I therefore believe RNAs, as well as peptides, may have had impact on RAFs in the early evolution of life. Moreover, at L72-74, the authors introduced a few examples of RAFs, but one of the works (17) is actually RNA-based.

Xavier et al.: We now reformulate our statement from “no reactions in KEGG have RNA catalysts assigned in Uniprot—an observation concerning the nature of catalysis in metabolism, not a bias in the sample.” to “no reactions in KEGG have RNA catalysts assigned in Uniprot, an observation concerning both the nature of catalysis in microbial metabolism and the limitations inherent to current databases” (lines 399-405). Even though several examples of RNA catalysis are known, these do not normally concern the small molecule metabolic networks that underpin genome-based approaches to metabolic maps. For example, tRNA transcript processing and the peptidyl transferase reaction are in KEGG but cannot be assigned RNA catalysts in Uniprot. There is room to move in that direction in future work, and that offers promise. We will pursue. But keeping the generality of the problem in focus, it is clear, and we underscored in the Discussion, that many of the cofactors are themselves RNA-related (White 1976). The difference is the polymerization state, which is then a comparatively small difference, all things considered. Nonetheless, the relationship between small molecule reactions and polymer chemistry that we uncover remains. We hope that the changes that we have introduced bring this point better into focus.

Referee: 3

Comments to the Author(s)

The manuscript deals with an extremely important problem: identify Reflexively Autocatalytic Food-generated networks (RAF), which are self-sustaining networks that collectively catalyze all their reactions and might be the precursors of more complex

molecular organizations. The existence of such objects would provide indirect evidence for a stepwise evolution leading to the origin of cells through processes of molecular self-organization.

The authors have analyzed various data sets to investigate different levels of ancient metabolism preserved in modern cells in order to identify RAF. Indeed they succeeded, to some extent, and the largest RAF that they have found comprises more than one thousand reactions in the whole prokaryotic anaerobic biochemical space.

The results of this manuscript are relevant for the various schools investigating the origin of life.

I should add that I am not an expert of the kind of statistical analysis that the authors have performed.

Xavier et al.: We thank R3 for such kind and gratifying words!!

In summary we thank the referees for their careful attention to our paper and we hope that the changes introduced will now allow the paper to go forward.

Appendix B

Associate Editor:

Board Member

Comments to Author:

The authors have responded to all reviewer comments/concerns, and edited the ms accordingly. Below are just a few minor suggestions for clarity.

Xavier et al.: We thank the editor for the quick and positive reply, and for the opportunity to clarify the minor points raised. Please find here short descriptions of the changes and their current locations in the tracked-changed manuscript where the edits are highlighted as requested (below).

Lines 70-72: sentence is unclear. Is there a typo?

Xavier et al.: Clarified (lines 70-72).

Lines 74-76: Sentence is a bit difficult to read.

Xavier et al.: Divided the sentence and clarified (lines 74-76).

Lines 203-205: Info. appears repetitive given info added in lines 128-130.

Xavier et al.: Thank you, removed fragment after comma (line 205).

lines 439-442: Language in parentheses makes sentence overly dense.

Xavier et al.: Thank you, removed fragment in parenthesis (line 432).